# Identification of SNPs Associated with Goose Meat Quality Traits Using a Genome-Wide Association Study Approach

**DOI:** 10.3390/ani13132089

**Published:** 2023-06-24

**Authors:** Guangliang Gao, Keshan Zhang, Ping Huang, Xianzhi Zhao, Qin Li, Youhui Xie, Chunhui Yin, Jing Li, Zhen Wang, Hang Zhong, Jiajia Xue, Zhuping Chen, Xianwen Wu, Qigui Wang

**Affiliations:** 1Chongqing Academy of Animal Science, Rongchang District, Chongqing 402460, China; guanglianggaocq@hotmail.com (G.G.); zhangkshlk1988@163.com (K.Z.);; 2Chongqing Engineering Research Center of Goose Genetic Improvement, Rongchang District, Chongqing 402460, China; 3Department of Laboratory Animal Sciences, Peking University Health Sciences Center, Beijing 100191, China

**Keywords:** goose, genome-wide association study, marker-assisted selection, meat traits, single nucleotide polymorphism

## Abstract

**Simple Summary:**

To screen and identify single nucleotide polymorphisms (SNPs) associated with geese meat quality traits, we performed genome-wide association study (GWAS) in a population of male Sichuan white geese (a Chinese local breed) at 70 days old to screen the SNPs in the goose genome. Additionally, genotypes of the selected SNPs associated with goose meat were detected using the matrix-assisted laser desorption ionization time-of-flight mass spectrometry (MALDI-TOP MS) method. The results revealed 43 SNPs that showed potentially significant associations with these traits. Additionally, we detected 28 annotated genes as potential candidate genes for the five meat quality traits. The enrichment analysis of genes within a 1Mb vicinity of SNPs showed potentially significant enrichment in the protein digestion and absorption pathway, as well as the Glycolipid metabolism pathway. The findings of this study offer novel genetic markers and candidate genes that can be used for marker-assisted selection of geese, laying the groundwork for understanding the genetic basis of goose meat quality traits and establishing a foundation for comprehending their genetic basis.

**Abstract:**

(1) Background: Goose meat is highly valued for its economic significance and vast market potential due to its desirable qualities, including a rich nutritional profile, tender texture, relatively low-fat content, and high levels of beneficial unsaturated fatty acids. However, there is an urgent need to improve goose breeding by identifying molecular markers associated with meat quality. (2) Methods: We evaluated meat quality traits, such as meat color, shear force (SF), cooking loss rate (CLR), and crude fat content (CFC), in a population of 215 male Sichuan white geese at 70 days of age. A GWAS was performed to identify potential molecular markers associated with goose meat quality. Furthermore, the selected SNPs linked to meat quality traits were genotyped using the MALDI-TOP MS method. (3) Results: A dataset of 2601.19 Gb of WGS data was obtained from 215 individuals, with an average sequencing depth of 10.89×. The GWAS revealed the identification of 43 potentially significant SNP markers associated with meat quality traits in the Sichuan white goose population. Additionally, 28 genes were identified as important candidate genes for meat quality. The gene enrichment analysis indicated a substantial enrichment of genes within a 1Mb vicinity of SNPs in both the protein digestion and absorption pathway and the Glycerolipid metabolism pathway. (4) Conclusion: This study provides valuable insights into the genetic and molecular mechanisms underlying goose meat quality traits, offering crucial references for molecular breeding in this field.

## 1. Introduction

Goose meat is a nutrient-rich meat that is considered a healthy food due to its high protein, low fat, and low cholesterol content, which is regarded as one of the ideal meat products and an important source of meat-based foods by the World Health Organization [1]. In 2021, Chinese goose meat production reached an impressive 570 million birds, accounting for 99% of Asia’s total and 94% of the global output. Furthermore, China achieved a goose meat yield of 1.77 million tons [2]. Renowned for its exceptional meat quality, the Sichuan white goose (Anser cygnoides domesticus) stands as a distinct local breed in China. It is characterized by a tender and delicate texture, contributing to its esteemed reputation and substantial market value. Over the past few years, there has been a steady rise in the market demand for goose meat [3,4]. This surge can be attributed to the increasing focus on healthy lifestyles, thereby highlighting significant opportunities within the goose meat industry. The meat quality of geese is a multifaceted attribute shaped by numerous genes, encompassing both major and minor effects. The intricate and multifaceted regulatory mechanism governing this trait adds to its complexity [5]. Previous studies have extensively documented the impact of factors such as breed, age, nutrition, environment, and disease on meat quality traits in geese [6]. Hence, identifying and screening molecular markers and key genes associated with these traits holds substantial potential to enhance breeding efficiency and elevate the overall quality of goose meat. Meat quality evaluation relies heavily on multiple meat indices, including lightness (L*), redness (a*), yellowness (b*) of meat color, pH value, shear force (SF), cooking loss rate (CLR), and crude fat content (CFC). Diets rich in methionine have been shown to positively influence pectoral muscle meat production, elevate pH value, enhance meat color, and reduce muscle shear [7,8,9]. The presence of fatty acids in the diet directly affects meat flavor and intramuscular fat production. Studies indicate that higher levels of soy phospholipids in the diet increase the fat content in pectoral and leg muscles, resulting in improved tethering force, reduced drip loss, and minimized cooking loss in the muscle [10,11]. Additionally, an increase in crude fat content (CFC) promotes intermuscular fat deposition, weakening the binding force between muscle fibers. As a result, shearing force is reduced, and meat tenderness is improved [12,13].

Genetic factors play a crucial role in determining meat quality traits alongside external factors. Genome-wide association studies (GWAS) have been utilized to investigate the relationship between phenotype and genotype in populations, yielding significant findings in the identification of genetic variations associated with meat quality traits in livestock and poultry [14,15,16,17,18]. The advancement of whole-genome resequencing technology enables accurate screening of various types of genetic variations, including single nucleotide polymorphisms (SNPs), insertion-deletion sites (InDels), structural variation sites (SVs), copy number variations (CNVs), presence/absence variations (PAVs), and others. Previous studies have identified candidate genes related to meat quality traits in chicken, broiler lines, and Pekin ducks [19,20]. For example, a GWAS focusing on chicken meat quality traits successfully identified genetic markers associated with characteristics such as meat color, pH, and water-holding capacity. Furthermore, it also suggested that *GJA1* may serve as a functional gene in the development of breast muscle in chickens [21]. Another study conducted on broiler lines identified seven potential candidate genes, including *SHH*, *LMBR1*, *IGF1R*, and *SLC16A*, which might be involved in controlling abdominal fat content. In the case of Pekin ducks, GWAS of body size and carcass traits revealed a specific mutation site located within the anti-lipogenesis gene *NR2F2* in high sebaceous Pekin ducks [22,23].

The utilization of whole-genome resequencing successfully detected single nucleotide polymorphisms (SNPs) associated with meat quality traits in the genome of Sichuan geese. Subsequent to that, a GWAS was conducted to precisely identify specific genomic regions and candidate genes associated with these traits. Moreover, we examined the frequency of SNPs associated with meat quality traits. The comprehensive findings of this study lay a solid foundation for future investigations into goose meat traits and make significant contributions to the advancement of genetic selection efforts in this field.

## 2. Materials and Methods

### 2.1. Experimental Animals and Phenotypic Traits

The male Sichuan white geese, selected as experimental animals from a shared incubation batch, were reared at the AnFu Waterfowl Breeding Base in Chongqing City, China (latitude: 105.478° N, longitude: 29.343° E). The geese were subjected to standardized conditions of diet, temperature, lighting, and unrestricted access to water during their rearing period. When the geese reached 70 days of age, a total of 215 healthy individuals were randomly chosen as experimental subjects. Blood samples were collected from the geese’s wings using vacuum tubes containing the anticoagulant EDTA, and these samples were subsequently stored at −20 °C to prepare them for future experiments. To measure the three meat color parameters (L*, a*, and b*) in three fixed points of 215 goose right pectoral muscle samples after 2 h of slaughter, an automatic colorimeter and a portable pH meter were used after calibration. The muscle samples were standardized for thickness and weight (M0) and then bathed in an 80 °C-thermostat water bath (model SW 22, Julabo GmbH, Seelbach, Germany). Once the center temperature of the samples reached 70 °C, they were incubated for a duration of 30 min. Subsequently, the samples were removed from the heat source and allowed to cool down to room temperature. The sample was then weighed (M1), and the cooking loss rate (CLR) was calculated as CLR = (M0 − M1)/M0 × 100%.

The cooled breast muscle was fixed perpendicular to the muscle fiber into long strips with a 1 cm × 1 cm cross-sectional area, and the strips were sheared with a tenderness meter (CLM-3), repeated 5 times to obtain the average shearing force (SF) of these samples. Fresh meat samples of 100g were freeze dried into powder, and the powder was weighed (Mp). The moisture content of freeze dried (MCFD) was calculated as MCFD = (100 − Mp)/100 × 100%. The fat was extracted by petroleum benzene from 1g of the freeze-dried powder, and the remaining material (M_R_) was dried at 105 °C for 2 h. The crude fat content (CFC) was then calculated as CFC = (1 − M_R_) × Mp/100 × 100%.

### 2.2. Whole-Genome Resequencing and SNP Calling

We extracted genomic DNA from whole blood samples of geese using the DP332 genomic DNA extraction kit from Tiangen Biotech. Subsequently, we evaluated the quality of the extracted DNA by measuring the OD260/280 ratio, which ranged from 1.8 to 1.89. To facilitate Illumina sequencing, the original DNA samples were diluted to a concentration of 50 ng/μL. We utilized 100 μL from each sample. Whole-genome sequencing (WGS) was performed using the Illumina HiSeq X Ten platform. Following the generation of WGS data, we performed quality control procedures using the NGS QC Toolkit software [24], which involved removing adapter sequences and residual primers and trimming low-quality bases. The filtered reads were aligned to the goose genome (version ASM1303099v1) using the BWA software [25,26], followed by SNP calling using GATK software [27]. The trait phenotype dataset in this study consisted only of individual-level samples with a genotype data completeness of over 95%. SNP filtering was performed using Plink [28] at the SNP level with the following parameters: -genome 0.1 –hwe 0.0000001 –maf 0.05 –mind 0.1. Additionally, individuals with less than 10% of missing genotypes were retained. As a result, a total of 203 individuals were included for further analysis. Afterward, we performed principal component analysis (PCA) analysis to evaluate goose population stratification using the model implemented in Plink based on the WGS data from 203 individuals.

### 2.3. GWAS

We utilized the GEMMA software [29] to conduct GWAS in order to identify single nucleotide polymorphisms (SNPs) associated with various goose meat quality traits, including CF, MCFD, CLR, SF, and meat color (L*, a*, b*). The GWAS employed a mixed linear model, expressed as the equation y = Wα + xβ + ε, where y represents the phenotypic value for all individuals. W, a covariance matrix (fixed effects: PC1-PC2 value), was used to control for population structure, while α is a vector of coefficients, including the intercept. The variable x corresponds to the genotype of the SNP or haplotype marker, and β represents the effect size of the SNP or haplotype marker on the phenotypes. Random residuals were represented by the vector ε. We assessed the significance of the associations between the SNPs and phenotypes using the Wald test statistic and applied Bonferroni’s correction to adjust the association analysis results. The significance and potential association thresholds in the whole-genome analysis were calculated using the formula *P* = 0.05/*N* or 1/*N*. Here, *P* represents the Bonferroni-corrected *p*-value for significant or potential associations across the entire genome, while *N* denotes the number of independent SNPs obtained from the population structure analysis. In this particular study, the determined thresholds for significant and potentially significant associations were 4.94 × 10^−9^ (0.05/10,072,006) and 1.0 × 10^−7^ (1/10,072,006), respectively. The Manhattan plots and quantile-quantile (Q-Q) plots were created using the “gap” package (Version 1.5-1, assess data 22 January 2023, https://cran.r-project.org/web/packages/gap/) [30] and “qqman” package (Version 0.1.8, assess data 19 April 2021, https://cran.r-project.org/web/packages/qqman/) [31] in the R project software. To evaluate the presence of false-positive signals in the obtained results, we utilized the GenABEL package and calculated the genomic inflation factor (k) [32].

To determine the genotypes of the SNPs significantly associated with meat quality traits in the Sichuan white goose population using GWAS, we designed amplification and extension primers (Table 1) to amplify the target sequences and facilitate the hybridization and elongation of the fragments at the specific nucleotide of interest. To assess the variation in meat quality traits among different genotypes of selected SNPs, we utilized a general linear model in JMP software (version 13.0). Following this, we computed the least-squares mean for each genotype and performed a Bonferroni test to evaluate the differences between the genotypes.

### 2.4. SNP Annotation

In this study, the BEDTools software [33] was used to extract genetic information from 1 Mb regions upstream and downstream of each potential SNP in the goose genome, while SNP annotation was conducted using Annovar software [34] (SnpEff, Annovar, VEP, Oncotator). Functional annotation analysis of the candidate genes was performed using the Metascape website [35] (https://metascape.org/, accessed on 12 December 2019).

## 3. Results

### 3.1. Phenotypic Description of the Goose Population

A fast and efficient fattening method is commonly used for raising meat geese. Generally, geese can be marketed after 70 days, and the meat quality of 70-day-old geese holds significant economic value in production practice. The statistical data of meat quality traits for the breast muscles of 70-day-old male Sichuan white geese are presented in Table 2 as part of this study, including crude fat content (CFC), meat color (L*, a*, and b*), shear force (SF), moisture content of freeze dried (MCFD), and cooking loss rate (CLR).

### 3.2. Whole-Genome Resequencing and SNP Calling

We obtained 2608.44 Gb of WGS data from 215 male Sichuan white geese, with an average depth coverage of 10.89×. After filtering, we obtained 2601.19 Gb of high-quality sequencing data, with an average mapping rate of 97.88% (ranging from 97.72% to 98.10%) when aligned to the reference genome sequence of geese. The average Q20 and Q30 values were 96.77% and 91.90%, respectively. The average GC content was calculated to be 43.63%. The subsequent analysis focused on 10,072,006 SNPs identified in the goose genome (version ASM1303099v1) (Figure 1).

### 3.3. The Goose Population Structure and the GWAS

The PCA analysis indicated that there was no significant population stratification (Figure 2). Hence, we employed a mixed linear model utilizing the GEMMA software package to conduct a genome-wide association study (GWAS) for the meat quality traits. We identified a total of 43 SNPs located within 28 genes (Table 3) that exhibited potentially significant associations with the meat quality traits (Figure 3).

### 3.4. The Genotypes of the Selected SNPs

We detected the 11 SNPs genotypes and compared the meat quality traits of different genotypes identified by GWAS (Table 4), including eight SNPs (chr13:25432644, ctg930:11921, chr36:5611038, ctg2092:176333, chr29:8278595, chr9:33911228, chr1:24867798, and chr7:16992800) are potentially significantly associated with b* meat yellowness and three SNPs (chr12:31782870, chr12:31781825, and chr2:36812553) are significantly associated with SF meat quality traits.

### 3.5. The Gene Annotation within 1 Mb of the Potentially Significant SNPs

We performed the gene annotation within 1Mb of the potentially significant SNPs with the goose meat quality traits by the KEGG database. The gene enrichment analysis shown that the genes were enrich in the Cushing syndrome (ko04934, *p* value = 1.80 × 10^−3^), human papillomavirus infection (ko05165, *p* value = 3.22 × 10^−3^ 0.003221264), ECM-receptor interaction (ko04512, *p* value = 3.53 × 10^−3^), protein digestion and absorption (ko04974, *p* value = 5.21 × 10^−3^), and Glycerolipid metabolism (ko00561, *p* value = 8.79 × 10^−3^) (Figure 4).

## 4. Discussion

The Sichuan white goose is a dual-purpose breed of Chinese poultry known for producing both meat and eggs. It is a valuable genetic resource and contributes significantly to the poultry industry, providing substantial economic and nutritional benefits in China. The quality of meat in the poultry industry is a crucial factor that affects consumer acceptance and, ultimately, the economic success of the industry. Poultry breeders use selective breeding and genetic improvement programs to enhance meat quality traits, aiming to produce birds that meet consumer preferences and demand. In this study, we investigated the meat quality traits of 70-day-old male Sichuan white geese. Using a GWAS, we screened and identified a total of 43 SNPs associated with CFC, CLR, meat color, and SF traits in a population of 215 Chinese local geese (Sichuan white goose).

The sample size is a critical factor in research design that can affect the validity and reliability of research findings [36]. A larger sample size can improve the results by reducing sampling error, increasing statistical power, increasing generalizability, and increasing precision. In this study, the principal component analysis shows that there is no significant stratification in the population (Figure 2), and the QQ plots indicated a good fit between the observed and predicted values, thereby suggesting the absence of any significant population stratification. In addition, we used MALDI-TOP MS to identify the 11 SNP sites associated with the meat quality SNP, as well as analyze the frequency for the candidate SNPs, which correspond to the results of the GWAS. To conclude, the results of this study are valid and reliable, and the SNPs can be used as a breeding marker for goose meat.

The crude fat content (CFC) of meat is a critical indicator of its quality, and it has been shown to be associated with several other quality traits. In this study, we investigated the genetic factors underlying CFC in geese and identified two potentially significant SNPs located in the transmembrane protein 65 (*TMEM65*) and mothers against decapentaplegic homolog 6 (*SMAD6*) genes. *TMEM65* is a mitochondrial inner-membrane protein that plays a crucial role in oxidative stress response, oxygen consumption, and citrate synthase activity [37,38,39]. As CFC is closely related to fatty anabolism, which is influenced by the oxidative function of mitochondria, we hypothesize that *TMEM65* may be involved in fat oxidation metabolism, indirectly affecting CFC levels. Fatty acid metabolism has a significant impact on the flavor and production of intramuscular fat, which, in turn, affects meat tenderness, water retention, and flavor presentation. Therefore, the identification of genetic markers associated with CFC, such as the identified SNP within the *TMEM65* in this study, can have important implications for the selection and breeding of animals with desirable meat quality traits.

Meat color (L*, a*, and b*) reflects changes in muscle physiology, biochemistry, and microbiology and is an important index for evaluating meat appearance. Meat color can impact the functional characteristics and quality of further processed meat products. Our study (Table 2) identified 30 SNPs potentially significantly associated with meat color (MC). Glutamine synthetase (*GSS*), one of the identified candidate genes, plays a crucial role in several cellular processes. Its main function is the production of glutathione, which serves to protect cells from oxidative damage and aids in amino acid transport. Additionally, GSS is involved in the detoxification of foreign compounds. The presence of GSS as a candidate gene suggests its significance in safeguarding cells and maintaining cellular homeostasis. Further exploration of GSS’s role in these processes would provide valuable insights into its contribution to cellular health and the potential impact on various biological functions [40]. Ras-related GTP-binding protein C (*RRAGC*) and solute carrier organic anion transporter family member 5A1 (*SLCO5A1*) play crucial roles in the cellular response to amino acid availability. *RRAGC*, along with three other Rag proteins, plays a crucial role in the activation of the mTORC1 pathway, which is sensitive to amino acid levels [41,42], which regulates metabolic and physiological processes such as lipid metabolism, autophagy, and muscle protein synthesis [43,44,45]. After slaughtering, oxygen supply to the cells is reduced, leading to the accelerated breakdown of adenosine triphosphate (ATP), increased free radicals, accumulation of lactic acid, and changes in pH. These redox reactions promote protein denaturation and increase fluid efflux, leading to changes in meat color and water-holding capability [46]. Then, the meat color and water-holding capability occurred to change. Therefore, we hypothesize that the *GSS*, *RRAGC*, and *SLCO5A1* may be responsible for the color presentation of the carcass by oxidative metabolism.

The shearing force (SF) is a critical parameter for assessing meat tenderness, which is influenced by various factors such as muscle fiber characteristics (e.g., diameter, density, type, and integrity), connective tissue composition (e.g., content, type, and cross-linking status), myogenic fibrin hydrolase properties, and intramuscular fat content. In this study, we identified seven SNPs located in Bcl-2-modifying factor (*BMF*), Nuclear envelope integral membrane protein 2 (*NEMP2*), and mitochondrial NAD-dependent malic enzyme (*NAD-ME*) that were potentially significantly associated with SF in goose. The correlations were further validated by the results of MALDI-TOF MS. In this study, we confirmed significant associations between the genotypes of SNP (chr2:36812553) in *BMF* and two SNPs (chr12:31782870, chr12:31781825) in *NAD-ME* with the shear force (SF) for geese meat. *BMF*, a member of the Bcl2 protein family, is a sensor for intracellular damage that can trigger cell apoptosis and potentially mediate muscle fiber atrophy [47,48]. *NAD-ME* participates in various metabolic pathways and provides reducing power NADPH for the synthesis of various cellular components [49]. In avian species, NAD-ME serves as the energy source for synthesizing long-chain fatty acids in poultry liver [50]. We hypothesize that *BMF* may directly impact muscle fiber structure, while *NAD-ME* may affect oxidoreductase activity and intramuscular fat generation in muscle, ultimately affecting SF.

There are three reasons why the SNPs from the GWAS were not previously reported. Firstly, in this study, we utilized the first chromosome-level genome sequence of geese (version ASM1303099v1) as the reference, which resulted in improved genome assembly and annotation compared to previous studies [25]. Furthermore, the absence of similar studies on data regarding goose meat qualities, along with the minimal changes undergone by geese during domestication from wild migratory geese, has resulted in the lack of reporting on numerous other genes related to meat quality in this study [51].

Metabolic pathways have comprehensive effects on meat quality traits [52], involving energy metabolism, nutrient utilization, metabolites, and oxidative status. Specifically, these pathways play a crucial role in regulating muscle pH and oxidative status, thus impacting meat color and water-holding capacity [53,54]. Through functional clustering analysis of annotated genes, we identified a group of candidate genes primarily clustered in metabolic pathways related to protein digestion and interaction (ko 04974) and glycerolipid metabolism (ko 00561). The content of glycerolipid directly influences the water-holding capability and tenderness of the meat. The glucose metabolism pathway affects the energy supply in muscle tissue [55], while the fatty acid metabolism pathway regulates fat accumulation and breakdown, thereby influencing the fat content and sensory characteristics of meat [56].

## 5. Conclusions

In this study, we performed GWAS with goose meat quality traits and identified 43 SNPs that demonstrated potentially significant associations with these traits. Additionally, we used the MALDI-TOP MS method to genotype the selected SNPs associated with the traits in the same population. Our findings revealed 28 annotated genes that show promise as potential candidate genes linked to the five investigated meat quality traits. These genes represent novel genetic markers and can be implemented in marker-assisted selection programs for geese. Importantly, our study contributes to a comprehensive understanding of the genetic mechanisms underlying meat quality traits in geese, establishing a solid foundation for future research in this field.

## Figures and Tables

**Figure 1 animals-13-02089-f001:**
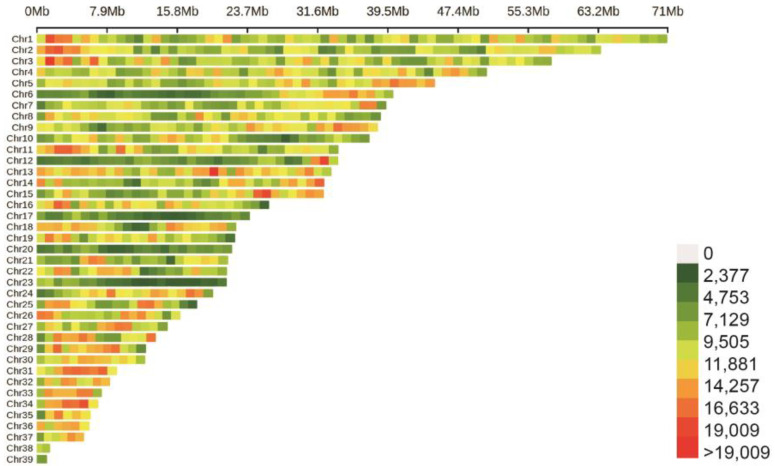
The SNP density in the goose genome within 1Mb window size.

**Figure 2 animals-13-02089-f002:**
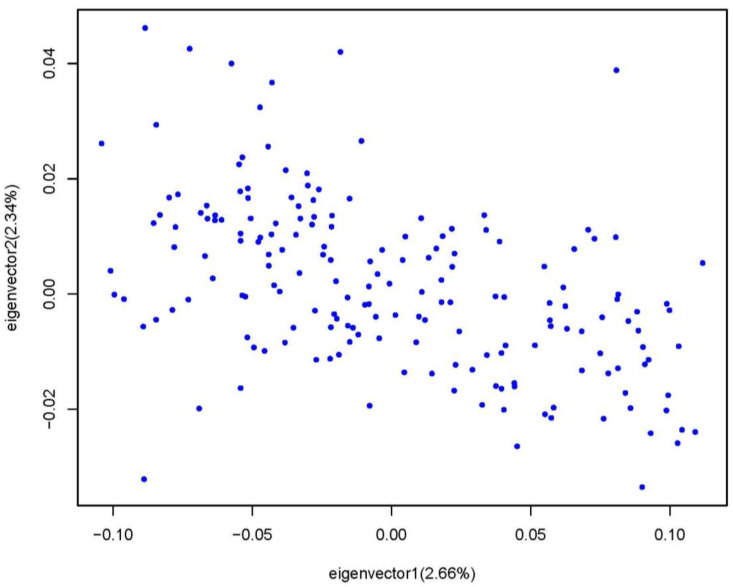
The results of PCA conducted on the goose population based on the whole genome SNPs. Note: The points in the plot represent individual members of the population.

**Figure 3 animals-13-02089-f003:**
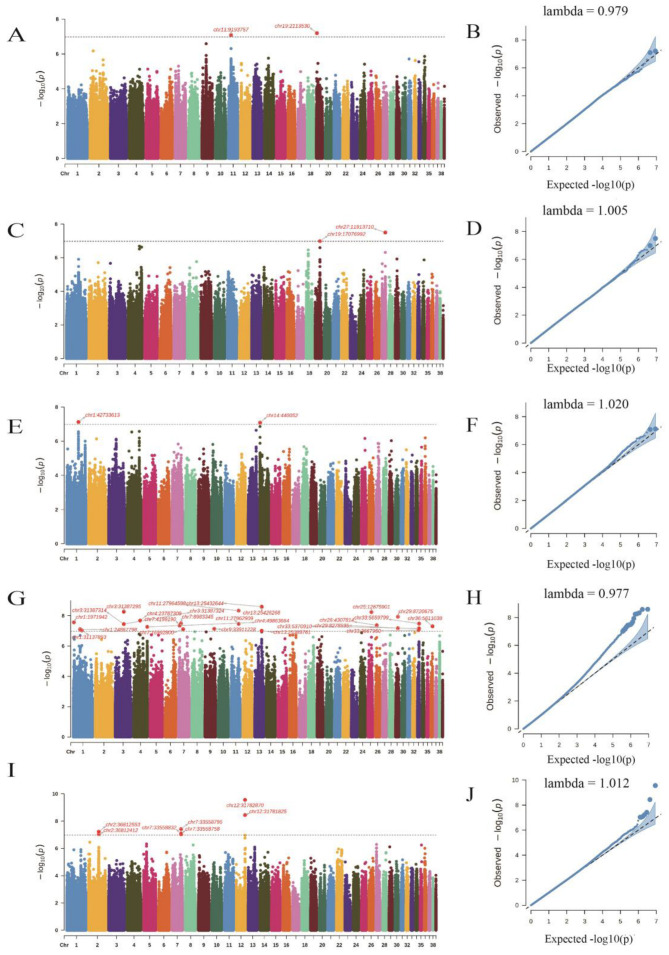
The Manhattan and Q-Q plots for different meat quality traits in male Sichuan white geese. Note: The plots are presented for the following traits: CFC (**A**,**B**); CLR (**C**,**D**); L* (meat lightness) (**E**,**F**); b* (meat yellowness) (**G**,**H**); SF (**I**,**J**). The threshold for significance is set at 1 × 10^−7^.

**Figure 4 animals-13-02089-f004:**
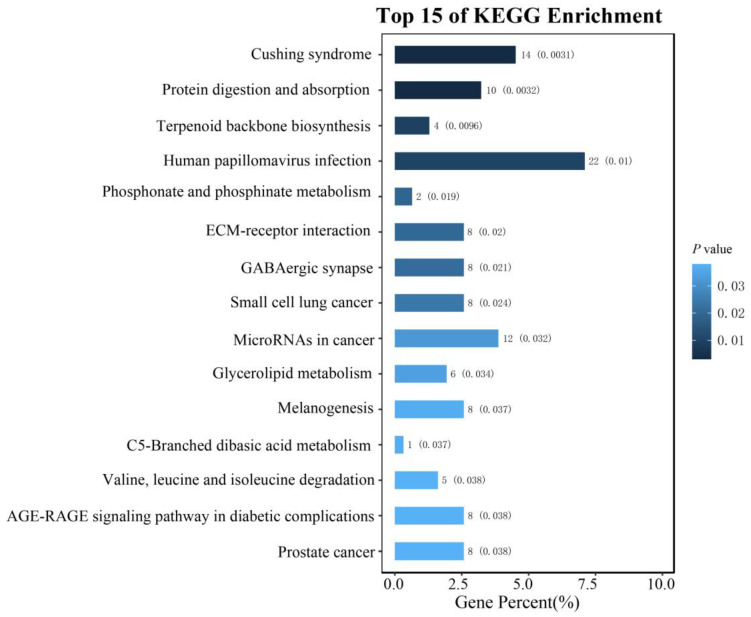
The functional analysis for the genes within 1 Mb within SNPs associated with goose meat quality traits.

**Table 1 animals-13-02089-t001:** The primer sequence of MALDI-TOF MS.

SNPs	2nd-PCRP	2nd-PCRP	UEP_SEQ
chr13:25432644	ACGTTGGATGGAAATACCCTGTTGTCTCCC	ACGTTGGATGACCTGTTTGACTCCTTTTGG	agaaCCTGTTGTCTCCCTCACTC
ctg930:11921	ACGTTGGATGTGCCACCGCAGGGATCACG	ACGTTGGATGTGGCAGCAGGGTGGGGAAA	GCCCCCTCCTGCACCTT
chr36:5611038	ACGTTGGATGTTCCCCCCTCGTTTGAATTG	ACGTTGGATGCCCAGTCTGAATTCCAACAC	tCGCTTTGATTTAGTTATTTTACTC
ctg2092:176333	ACGTTGGATGGGAATATGTAGACTACGTCTG	ACGTTGGATGTTTTGGACAAACAGGAGACC	cccaTAGACTACGTCTGCCATCT
chr29:8278595	ACGTTGGATGAAGGATTTGGGAAGCAGGAAC	ACGTTGGATGTGCACCGGGGAGGAGGAGA	CAGGAACCGAGGGAAATGC
chr9:33911228	ACGTTGGATGCCTGGAGGCAATCAAAGATG	ACGTTGGATGCTTAAGTCGCCTTGGTACAC	CAGGAGTTAAGGGAGAAAAT
chr1:24867798	ACGTTGGATGAGCCTGATGCAGTCACATCC	ACGTTGGATGGTGTGTGCCAGACAAAACTC	GCTGCCTGCCCAGAACT
chr7:16992800	ACGTTGGATGTCACATTGGCAGGGTCCAAC	ACGTTGGATGTTATAGTCTGCTCTGGACTG	gtCAGGGTCCAACTCAGTCTCC
chr12:31782870	ACGTTGGATGACAAAGAACACATCGCAAGG	ACGTTGGATGGGCAGCAGCTTTCAGCAAAC	CAATAATGTTTAACGTTAGACTC
chr12:31781825	ACGTTGGATGTTCTGCAACGCTCGAAATCC	ACGTTGGATGTTTAACCTACGCATGCCTCC	CCAAAGACCTGTTGAGA
chr2:36812553	ACGTTGGATGGTGGAAGAAGACATCACTGG	ACGTTGGATGGCAGGAGAAAAAAGCATAAG	tcGATCTTACTTTTTTATCTTCCATTA

**Table 2 animals-13-02089-t002:** Descriptive statistics were calculated for the meat quality traits of male Sichuan white geese at 70 days of age.

Traits	Number	Mean	STDV	Minimum	Maximum	CV (%)
CFC	199	9.92	1.89	4.69	15.52	0.19
MCFD	199	75.07	0.94	72.26	77.54	0.01
CLR	194	13.13	3.01	4.66	22.36	0.23
L* (meat lightness)	205	23.6	3.65	15.53	37.29	0.15
a* (meat redness)	205	48.77	3.02	39.76	56.06	0.06
b* (meat yellowness)	205	19.5	2.32	13.05	30.8	0.12
SF (kgf)	197	3.81	0.83	1.66	5.82	0.22

Note: CFC: crude fat content; MCFD: moisture content of freeze dried; CLR: cooking loss rate; L*: meat lightness; a*: meat redness); b*: meat yellowness; SF (kgf): shear force.

**Table 3 animals-13-02089-t003:** Summary of the 43 snp markers potentially associated with five meat quality traits in male Sichuan white geese.

SNP	Chr	Position (bp)	Allele1	Traits	*p* Value	Gene
chr11:9193757	chr11	9193757	G/A	CFC	8.12 × 10^−8^	*TMEM65*
chr19:2113530	chr19	2113530	C/T	CFC	6.34 × 10^−8^	*SMAD6*
chr19:17076992	chr19	17076992	T/G	CLR	1.03 × 10^−7^	*SYTC2*
chr27:11913710	chr27	11913710	G/A	CLR	3.16 × 10^−8^	*MSI1H*
chr1:42733613	chr1	42733613	T/C	L* (meat lightness)	7.56 × 10^−8^	*LIN1*
chr14:440052	chr14	440052	T/C	L* (meat lightness)	8.15 × 10^−8^	*GSS*
chr1:1971942	chr1	1971942	C/T	b* (meat yellowness)	2.74 × 10^−8^	*EXTL3*
chr1:24867798	chr1	24867798	C/G	b* (meat yellowness)	7.82 × 10^−8^	*PRSS55*
chr1:31137893	chr1	31137893	C/T	b* (meat yellowness)	9.31 × 10^−8^	*PTP4A1*
chr3:31387295	chr3	31387295	G/A	b* (meat yellowness)	5.43 × 10^−9^	*TMEM19*
chr3:31387314	chr3	31387314	G/A	b* (meat yellowness)	3.57 × 10^−8^	*TMEM19*
chr3:31387324	chr3	31387324	A/G	b* (meat yellowness)	3.57 × 10^−8^	*TMEM19*
chr7:8983345	chr7	8983345	A/G	b* (meat yellowness)	3.20 × 10^−8^	*ACKR3*
chr7:16992800	chr7	16992800	C/T	b* (meat yellowness)	7.84 × 10^−8^	*CHIN*
chr7:4199190	chr7	4199190	A/G	b* (meat yellowness)	4.47 × 10^−8^	*GULP1*
chr9:33911228	chr9	33911228	A/G	b* (meat yellowness)	7.35 × 10^−8^	*DOCK1*
chr11:27962909	chr11	27962909	A/G	b* (meat yellowness)	3.43 × 10^−8^	*PP4R1*
chr11:27964588	chr11	27964588	G/T	b* (meat yellowness)	4.66 × 10^−9^	*SLCO5A1*
chr13:25432644	chr13	25432644	C/T	b* (meat yellowness)	2.55 × 10^−9^	*CATC*
chr13:25426268	chr13	25426268	G/A	b* (meat yellowness)	2.62 × 10^−9^	*CATC*
chr13:25389761	chr13	25389761	T/G	b* (meat yellowness)	9.87 × 10^−8^	*CATC*
chr26:4307814	chr26	4307814	T/C	b* (meat yellowness)	4.26 × 10^−8^	*PP4R1*
chr29:8720675	chr29	8720675	T/C	b* (meat yellowness)	1.16 × 10^−8^	*MRPS2*
chr33:5370910	chr33	5370910	A/C	b* (meat yellowness)	6.90 × 10^−8^	*NO40*
chr33:5659799	chr33	5659799	A/C	b* (meat yellowness)	3.33 × 10^−8^	*PUM1*
chr33:3667950	chr33	3667950	C/T	b* (meat yellowness)	8.99 × 10^−8^	*RRAGC*
chr36:5611038	chr36	5611038	G/T	b* (meat yellowness)	4.99 × 10^−8^	*GOGA7*
ctg2092:176333	ctg2092	176333	C/T	b* (meat yellowness)	6.16 × 10^−8^	*RXRA*
ctg745:127150	ctg745	127150	A/G	b* (meat yellowness)	5.53 × 10^−8^	*NCOA2*
chr4:23787309	chr4	23787309	A/G	b* (meat yellowness)	2.08 × 10^−8^	*None*
chr4:49863664	chr4	49863664	T/A	b* (meat yellowness)	5.49 × 10^−8^	*None*
chr25:12875901	chr25	12875901	A/G	b* (meat yellowness)	5.77 × 10^−9^	*None*
chr29:8278595	chr29	8278595	C/T	b* (meat yellowness)	6.64 × 10^−8^	*None*
ctg834:24329	ctg834	24329	C/T	b* (meat yellowness)	5.15 × 10^−8^	*None*
ctg930:11921	ctg930	11921	G/A	b* (meat yellowness)	5.30 × 10^−9^	*None*
ctg956:92804	ctg956	92804	C/T	b* (meat yellowness)	2.73 × 10^−8^	*None*
chr2:36812553	chr2	36812553	C/T	SF (kgf)	5.94 × 10^−8^	*BMF*
chr2:36812412	chr2	36812412	G/C	SF (kgf)	9.22 × 10^−8^	*BMF*
chr7:33558795	chr7	33558795	A/T	SF (kgf)	3.84 × 10^−8^	*NEMP2*
chr7:33558758	chr7	33558758	C/A	SF (kgf)	8.04 × 10^−8^	*NEMP2*
chr7:33558832	chr7	33558832	T/C	SF (kgf)	9.30 × 10^−8^	*NEMP2*
chr12:31782870	chr12	31782870	A/G	SF (kgf)	2.78 × 10^−10^	*NAD-ME*
chr12:31781825	chr12	31781825	G/A	SF (kgf)	3.62 × 10^−9^	NAD-ME

Note: The threshold for significance significant and potentially significant association were set at 4.94 × 10^−9^ and 1 × 10^−7^ respectively. *ACKR3*: Atypical chemokine receptor 3; *BMF*: Bcl-2-modifying factor; *CATC*: Dipeptidyl peptidase 1; *CHIN*: N-chimaerin; *DOCK1*: Dedicator of cytokinesis protein 1; *EXTL3*: Exostosin-like 3; *GOGA7*: Golgin subfamily A member 7; *GSS*: Glutamine synthetase; *GULP1*: PTB domain-containing engulfment adapter protein 1; *LIN1*: LINE-1 reverse transcriptase homolog; *MRPS2*: 28S ribosomal protein S2, mitochondrial; *MSI1H*: RNA-binding protein Musashi homolog 1; *NAD-ME*: NAD-dependent malic enzyme, mitochondrial; *NCOA2*: Nuclear receptor coactivator 2; *NEMP2*: Nuclear envelope integral membrane protein 2; *NO40*: Nucleolar protein of 40 kDa; *PP4R1*: Serine/threonine-protein phosphatase 4 regulatory subunit 1; *PRSS55*:Serine protease 55; *PTP4A1*: Protein tyrosine phosphatase type IVA 1; *PUM1*: Pumilio homolog 1; *RRAGC*: Ras-related GTP-binding protein C; *RXRA*: Retinoic acid receptor RXR-alpha; *SLCO5A1*: Solute carrier organic anion transporter family member 5A1; *SMAD6*: Mothers against decapentaplegic homolog 6; SYTC2: Probable threonine--tRNA ligase 2, cytoplasmic; *TMEM19*: Transmembrane protein 19; *TMEM65*: Transmembrane protein 65; None: No annotated genes were found in the specified region of the genome. L*: meat lightness; b*: meat yellowness.

**Table 4 animals-13-02089-t004:** The frequency and genotypes of the selected SNPs associated with the goose meat quality.

Traits	Number	SNPs	Genotypes (Frequency %)		
b* (meat yellowness)	203	chr13:25432644	CC	CT	TT
			17.61 ± 2.07 ^b^ (1.97)	17.56 ± 2.92 ^b^ (0.66)	19.31 ± 0.25 ^a^ (97.37)
b* (meat yellowness)	203	ctg930:11921	AA	GA	GG
			19.18 ± 0.23 ^b^ (72.37)	20.28 ± 0.41 ^a^ (23.03)	19.03 ± 0.88 ^b^ (4.61)
b* (meat yellowness)	203	chr36:5611038	GG	GT	TT
			19.83 ± 2.76 ^a^ (0.58)	17.71 ± 1.04 ^b^ (4.62)	19.44 ± 0.22 ^a^ (94.80)
b* (meat yellowness)	203	ctg2092:176333	CC	TC	TT
			18.04 ± 1.15 ^b^ (3.49)	19.24 ± 0.45 ^a^ (13.56)	19.48 ± 0.26 ^a^ (70.93)
b* (meat yellowness)	203	chr29:8278595	CC	CT	TT
			17.99 ± 2.80 ^b^ (0.56)	19.21 ± 0.60 ^a^ (13.56)	19.39 ± 0.23 ^a^ (85.88)
b* (meat yellowness)	203	chr9:33911228	AA	GA	GG
			20.32 ± 1.62 ^a^ (2.27)	19.52 ± 0.75 ^b^ (8.52)	19.31 ± 0.23 ^b^ (89.20)
b* (meat yellowness)	203	chr1:24867798	CC	GC	GG
			16.33 ± 2.79 ^b^ (0.57)	19.08 ± 0.72 ^a^ (9.09)	19.41 ± 0.23 ^a^ (90.34)
b* (meat yellowness)	203	chr7:16992800	CC	CT	TT
			15.43 ± 2.77 ^b^ (1.12)	19.30 ± 0.57 ^a^ (13.48)	19.40 ± 0.23 ^a^ (85.39)
SF	203	chr12:31782870	AA	GA	GG
			3.12 ± 0.97 ^b^ (1.12)	3.82 ± 0.25 ^a^ (8.43)	3.74 ± 0.08 ^a^ (90.45)
SF	203	chr12:31781825	AA	GA	GG
			3.72 ± 0.08 ^b^ (87.01)	3.89 ± 0.22 ^a^ (12.99)	\
SF	203	chr2:36812553	CC	CT	TT
			\	3.60 ± 0.22 ^b^ (15.49)	3.73 ± 0.09 ^a^ (84.51)

Note: Different letters in the same row indicate a significant difference between the Genotypes (*p* < 0.05).

## Data Availability

The whole-genome sequencing (WGS) data have been deposited in the GenBank databases under the accession number PRJNA595357. The data can be accessed at https://www.ncbi.nlm.nih.gov/bioproject/PRJNA595357, accessed on 12 December 2019.

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
