# Peer review of "Identification of SNPs Associated with Goose Meat Quality Traits Using a Genome-Wide Association Study Approach"

_animals, 2023, doi:10.3390/ani13132089_

Round 1

Reviewer 1 Report

The manuscript is interesting, well writing and presented.

There are some questions and suggestions for better presenting the results.

1-Is there pedigree information for the population? If yes, it should be included in the model. It will permit a better estimation of the effects of the markers. Moreover, it would permit to estimate the variance explained by each marker (that was not calculated nor presented).

2-If pedigree is not available, it would be better to proceed the GWAS using the haplotype block. They took in consideration the linkage across markers and better reflect the significance and variance explained.

The authors said it was done, but the results were not presented.

Author Response

Comment 1:

The manuscript is interesting, well writing and presented.

There are some questions and suggestions for better presenting the results.

1-Is there pedigree information for the population? If yes, it should be included in the model. It will permit a better estimation of the effects of the markers. Moreover, it would permit to estimate the variance explained by each marker (that was not calculated nor presented).

Response 1:

In this study, pedigree information for the population was not available, so pedigree information was not included in the GWAS analysis model. We will collect complete pedigree information for future studies and incorporate pedigree information in subsequent GWAS research.

Comment 2:

2-If pedigree is not available, it would be better to proceed the GWAS using the haplotype block. They took in consideration the linkage across markers and better reflect the significance and variance explained.

The authors said it was done, but the results were not presented.

Response 2:

Thank you for the clarification. We have rectified the inaccuracies in the main text regarding the inclusion of GWAS analysis using the haplotype block. Additionally, it has been clarified that the SNPs associated with the five goose meat quality traits were not found within the haplotype block. These revisions have been made to ensure the accuracy of the information presented in the paper.

Reviewer 2 Report

To Authors, 

This research is useful and can be used as a good academic reference. However, there are some questions need to be completed.

Lines 39-41: You should provide statistics on production volumes and growth trends in the global goose meat production industry. In addition, you should add information about the Sichuan white geese in the introduction section  as well.

Line 93: What are your criteria for selecting geese for this study? Please add more information to the script menu.

Line 184: In Table 1, please list the full names of CF, MCFD, CLR, L* a* b*, and SF below the table.

In the discussion section, you should add more information and knowledge about the SNPs discovered; for example, you should provide a rationale for why the SNPs identified in this study are these. Or why SNPs located in this population might not be found in other populations?

Best Regards,

Reviewer

Author Response

Comment 1:

This research is useful and can be used as a good academic reference. However, there are some questions need to be completed.

Lines 39-41: You should provide statistics on production volumes and growth trends in the global goose meat production industry. In addition, you should add information about the Sichuan white geese in the introduction section as well.

Response 1:

We have provided detailed information on goose production volumes and growth trends in the global meat production industry. Additionally, we have included information about Sichuan white geese in both the introduction section and the main text (lines 50-56). “In 2021, Chinese goose meat production reached an impressive 570 million birds, ac-counting for 99% of Asia's total and 94% of the global output. Furthermore, China achieved a goose meat yield of 1.77 million tons. Renowned for its exceptional meat quality, the Sichuan white goose (Anser cygnoides domesticus) stands as a distinct local breed in China. It is characterized by a tender and delicate texture, contributing to its esteemed reputation and substantial market value. Over the past few years, there has been a steady rise in the market demand for goose meat.”

Comment 2:

Line 93: What are your criteria for selecting geese for this study? Please add more information to the script menu.

Response 2:

In this study, a total of 215 healthy individuals were randomly chosen as experimental subjects, instead of being selected based on specific criteria.

Comment 3:

Line 184: In Table 1, please list the full names of CF, MCFD, CLR, L* a* b*, and SF below the table.

 Response 3:

Thanks for incorporating the advice. We have provided the full names for the traits.

Comment 4:

In the discussion section, you should add more information and knowledge about the SNPs discovered; for example, you should provide a rationale for why the SNPs identified in this study are these. Or why SNPs located in this population might not be found in other populations?

Response 4:

Thanks for the valuable advice. There are three reasons why the SNPs from the GWAS analysis were not previously reported. Firstly, in this study, we utilized the first chromosome-level genome sequence of geese (version ASM1303099v1) as the reference, which resulted in im-proved genome assembly and annotation compared to previous studies. Further-more, the absence of similar studies on data regarding goose meat qualities, along with the minimal changes undergone by geese during domestication from wild migratory geese, has resulted in the lack of reporting on numerous other genes related to meat quality in this study. We had added the sentence in the main text as following (line 332-338).

Reviewer 3 Report

attached

Need to be improved.

Author Response

Comment 1:

Abstract

1 line (ln) 15, please add the function category of these SNPs

Response 1:

We appreciate the valuable advice and have included the gene enrichment analysis for genes within a 1 Mb vicinity (1 Mb upstream and 1 Mb downstream) in our study (Main text, line 21-22) as following “The enrichment analysis of genes within a 1Mb vicinity of SNPs showed a significant enrichment in the Protein digestion and absorption pathway, as well as the Glycolipid metabolism pathway”.

Comment 2:

2 Same for the 28 genes in ln 28

Response 2:

Thanks for the valuable advice. We had added the gene enrichment analysis for the genes within 1 Mb (upstream 1 Mb and downstream 1Mb) (Main text, line 38-40) as following “The gene enrichment analysis indicated a substantial enrichment of genes within a 1Mb vicinity of SNPs in both the Protein digestion and absorption pathway and the Glycolipid metabolism pathway.”

Comment 3:

Introduction

1 ln 39-40, missing supporting numbers for the increasing demands.

Response 3:

We appreciate the valuable advice and have provided the reference to support the statement (Main text, line 55-56) as following “Over the past few years, there has been a steady rise in the market demand for goose meat”.

Comment 4:

Methods

1 ln 93, what is the gender of these geese? Are they evenly distributed? Also, what is the species name of Sichuan white geese?

Response 4: 1

Thanks for the correction. In this study, we selected the male Sichuan white geese as experimental animals. The species name of Sichuan white geese is Anser cygnoides domesticus.

Comment 5:

2 ln 129, section 2.3, please add the number of samples have been filtered out and kept accordingly.

Response 5:

Thanks for the valuable advice. There are 203 individuals were included for further analysis, we had provided the information in line 152 of main text as following “A total of 203 individuals were included for further analysis”.

Comment 6:

3 ln 140, if these geese have a random population structure, please add random selection method in section 2.1.

Response 6:

Thanks for the valuable advice. When the geese reached 70 days of age, a total of 215 healthy individuals were randomly chosen as experimental subjects (Main text, line 116 -117).

Comment 7:

4 ln 175, where were the genome data stored? Please upload all the sequencing data to a publicly available online site.

Response 7:

All of the 215 whole genome resequencing data stored in SRA, the BioProject number is PRJNA595357.

Comment 8:

Results

1 ln 194, please clarify what is 1Mb window size. Also, why there were only 39 chromosomes?

Response 8:

In this study, In this study, we utilized the BEDTools software was used to extract genetic information from 1 Mb regions upstream and downstream of each potential SNP in the goose genome, while SNP annotation was conducted using Annovar software, we had revised the sentence in main text of line 182-184. In goose genome, there are 39 (n=78) pair chromosomes in goose genome.

Comment 9:

2 ln 205, will it make more sense if add function category then arrange this table based on the category within each trait?

Response 9:

Thanks for the valuable advice. In fact, Due to the limited research conducted on geese and the scarcity of gene functional annotations, we have undertaken the task of functionally annotating all 43 single nucleotide polymorphisms (SNPs) identified from the genome-wide association analysis. Specifically, our focus lies on the 1 Mb genomic regions upstream and downstream of these SNPs.

We had provided the gene enrichment analysis for the gene annotation within 1Mb of the significantly SNPs in line 245-253,

Comment 10:

3 ln 210, Fig.3. the red letters are hard to see and need to be included in the legend.

Response 10:

Thanks for the valuable advice. The red letters in Figure 3 were provide in the table 2 (main text, line 215-216)

Comment 11:

Discussion

ln 250-251, have these two genes been reported before? Also, for the rest discussion, how the findings from this study were related with other studies?

Response 11:

There are three reasons why the SNPs from the GWAS analysis were not previously reported. Firstly, in this study, we utilized the first chromosome-level genome sequence of geese (version ASM1303099v1) as the reference, which resulted in im-proved genome assembly and annotation compared to previous studies. Furthermore, the absence of similar studies on data regarding goose meat qualities, along with the minimal changes undergone by geese during domestication from wild migratory geese, has resulted in the lack of reporting on numerous other genes related to meat quality in this study.  

Comment 12:

Lastly, the authors need to correct some grammar and format issues (e.g., Table 1, content in columns Traits, Number, STDV, and CV% were not centered).

Response 12:

Thanks for the valuable advice. We had improved the academic style, spelling, grammar, clarity, concision, and overall readability of the paper, including the Table1.

Round 2

Reviewer 1 Report

Manuscript accepted to publication

Author Response

Thank you

Reviewer 2 Report

Dear Authors,

I am satisfied with the author's edits and consider them sufficient for publication.

Best Regards

Reviewer

Author Response

Thank you